# How Does a Pandemic Disrupt the Benefits of eCommerce? A Case Study of Small and Medium Enterprises in the US

**Raziel Bravo [1], Mario Gonzalez Segura [2,*], Olawale Temowo [1] and Subhashish Samaddar [3]**

1   J. Mack Robinson College of Business, Georgia State University, Atlanta, GA 30326, USA;
    rbravo1@student.gsu.edu (R.B.); otemowo1@student.gsu.edu (O.T.)
2   Public Service Outreach, J. Mack Robinson College of Business, Georgia State University,
    Atlanta, GA 30326, USA
3   Department of Management, J. Mack Robinson College of Business, Georgia State University,
    Atlanta, GA 30326, USA; s-samaddar@gsu.edu
*   Correspondence: mgonzalezsegura1@student.gsu.edu

**Abstract:** Inspired by the ongoing disruption to businesses across the world, this research focuses on how the COVID-19 pandemic has affected the contribution of eCommerce to small and medium-sized enterprises (SMEs). Our study seeks to establish an eCommerce-driven response to this natural disruption, by asking the questions; How do eCommerce platforms impact SMEs? How does eCommerce affect an SME's three major business functions during a global disruption? We employ a qualitative case study method, using interviews as our primary data source, along with secondary data from industry and company records. We discuss these case studies through the framework of the actor network theory (ANT), identifying eCommerce and other platforms that SMEs use as actors in their network. We interviewed eight SMEs involved in the physical sale and distribution of consumer goods, each of which had been operating for at least two years and had a maximum of 70 employees. On average, we found that 44% of the SMEs in this study benefitted from using eCommerce in key business areas, with 46% improving their operations, 47% improving sales and marketing, and 39% improving finance. We also found that SME adoption of eCommerce during the pandemic grew in response to these benefits. Of the eight companies we studied, four had begun developing full eCommerce operations and three more planned to develop them as the global situation further normalizes.

**Keywords:** eCommerce; SME; COVID-19; pandemic; disruption

## 1. Introduction

Globalization has created an increasing interdependence among countries and removed boundaries in domestic and international markets, offering businesses opportunities to expand their profiles abroad. Several factors have contributed to these opportunities, including that geographic and industrial specialization has created a world in which specific products are often best sourced from specific locations [1]. Further, the cost of shipping goods internationally has fallen to levels that can be absorbed by pricing products for the destination markets [2].

The effect of globalization on businesses—and particularly on smaller businesses—has been well studied in academia [3] because smaller organizations make a significant contribution to trade. In fact, the rise of smaller businesses has been an essential trend since the 1990s, when they became engines for economic development [4]. Now, SMEs across the world are increasingly relying on eCommerce channels to create and capture value [5]. The growth in eCommerce operations is being fueled by several global and technological developments. Internet channels have facilitated communications that allow even the smallest of businesses to run global operations [6]. The growing popularity of global value chains and the increased reliability of long supply chains have further encouraged small

businesses to transact with suppliers and customers across the world, and that customer base is exploding.

In the year 2000, there were only 413 million internet users in the world; by 2016, the number of users had increased to 3.4 billion [7]. Internet users translate to consumers for most businesses. Although technology allowed SMEs access to these consumers, they must compete for their business in a global environment, with bigger, better funded, and more experienced players. eCommerce helps SMEs meet this challenge [8] by allowing them to expand their geographical scope, develop a larger customer base in new markets, and improve their products for customer satisfaction.

Up and downstream, businesses are increasingly digitizing their operations to tap into these platforms. In response, upstream operations are consolidating to take advantage of economies of scale [9]. Meanwhile, downstream businesses contiguous to the customer remain highly fragmented, especially in emerging markets. Although clearly providing benefits, these channels increasingly expose SMEs to supply chain risks, especially in the face of natural disruptions [10] and other disruptive forces beyond their control.

As our study found, all case study participants retained manual processes for ordering from overseas suppliers for inventory replenishment due to the lack of infrastructure, processes, and resources to digitize these upstream processes. About half of the interview participants relied on conventional means of using email, phone calls, and text messaging to place orders from their suppliers. They heavily rely on supplier relationships. During the recent pandemic, these SMEs claimed that they maintained enough inventory in stock—for the reason that orders significantly reduced.

Among the cases included in our study, 50% of the respondents were driven to change supply chain processes during the COVID-19 pandemic. To minimize the effect of the delays during this disruption, 100% of the SMEs who took part in our research relied on their logistics partners such as fulfillment centers, courier companies, and payment processing companies for downstream processes to meet customer expectations.

Long-term relationships with other supply chain partners including logistics providers, customs brokers, and truckers supported their transportation requirements to replenish inventory despite delays caused by equipment and capacity shortages. Despite this workaround, destination-controlled resources at the ports created another issue. The shortfall of manpower at freight terminals, truck drivers, and warehouse workers created another domino effect of delays in handling inbound inventory replenishment.

While platforms of contracted partners provided visibility to their customers, the lack of resources among these intermediaries also disabled real-time and accurate updates that impacted their revenues.

In the following, we describe the complicated relationship between eCommerce and SMEs by exploring how global disruptions related to the COVID-19 pandemic impacted the eCommerce functions at eight SMEs in the United States. First, however, we offer an overview of the study context and examine existing literature to identify the gaps in available knowledge on the subject.

We will discuss our findings within the framework of the Actor-Network Theory. This theory, developed in the 1980s, proposes that all interactions are mediated through objects. In our research, these actors include the eCommerce website, the process-supporting platforms, third-party vendors, and SME employees. As the theory states, combining all actors creates a heterogeneous group with aligned interests, where one actor—usually the SME business owner—is indispensable in decision making. To ensure a stable network, this actor also mediates and unites all other actors in the network that are critical to the process. This unification occurs by developing relationships and emphasizing shared common interest. The business owners typically create these alliances and networks to ensure that they can successfully meet customer demands.

eCommerce allows SMEs to benefit from globalization by expanding their ability to access customers around the world and across customer demographics and categories. This emergence of a new range of markets has made strategic flexibility essential for SMEs,

as such expansion requires a consistent rethinking and adapting to continually changing dynamics in a global environment [11].

SMEs are typically defined based on the number of employees, annual sales, and ownership for most countries. The most frequent upper limit on employees is 250—as in the European Union—but some countries set the limit at 200 employees while others, such as the United States, set it higher at 500 employees.

For globalization to work to their advantage, small SMEs unbundle through fragmentation, offshoring or vertical specialization of the value chain [12]. By unbundling operations that do not directly contribute to their core competencies [13], SMEs can focus on these core competencies to maintain their competitiveness. This process of unbundling is more advantageous to SMEs since they generally have limited resources.

Our study's goal is to evaluate the impact of disruptions on eCommerce. Here, disruptions refer to any event or innovation within an industry that radically and permanently changes how all companies in that industry operate [14]. Natural disruptions include fires, floods, severe storms, diseases, and other events created by nature, while disruptions directly created by humans include terrorism, strikes, regulatory changes, and technological challenges. The latter are regular occurrences for many organizations, particularly those companies dealing with global supply chain processes. Such human-created disruptions have become increasingly common due to the increase in global sourcing activities over the past 30 years [15]. Such disruptions are inevitable and can have a cumulative negative—and typically long-term—impact on an organization's performance [16]; this is particularly true for SMEs, and hinders the scalability of their business models.

The SMEs focused on keeping afloat during the recent global pandemic. They performed operations and finance functions at the same levels to focus on sales and marketing initiatives. Only two out of the eight cases made drastic changes by immediately implementing eCommerce due to the closure of brick-and-mortar stores. Scarce resources to manage the new operations, however, limited their ability to improve other processes.

We apply the Actor-Network Theory (ANT) as a framework for our research, identifying human and non-human actors and their relations with each other in an SMEs eCommerce network. This network consists of an SME organization and its employees, the supporting eCommerce infrastructure, external systems (both integrated and disintegrated with the eCommerce platform), suppliers, vendors, and customers. The coordination of efforts among actors in an SMEs eCommerce-related network enables effective processes, while each actor in it adjusts and reacts in a fluid way. The pandemic is the ANT's "social force" and, as our findings show, it stimulated the actors to adjust behaviors.

While eCommerce platforms have had an overall positive impact on global economic growth during such disruptions, specific case studies of individual eCommerce businesses and SMEs are likely to reveal adverse effects as well. We therefore set out to examine the key SME business functions and answer two research questions: (1) *How do eCommerce platforms impact SMEs?* (2) *How does eCommerce affect an SME's three major business functions during a global disruption*?

## 2. Literature Review

### 2.1. The Contributions of eCommerce to SMEs

SMEs can use eCommerce to enhance and expand their reach. Consider, for example, The Natural Baby Company in Bozeman, Montana. This SME deployed different digital tools to grow its operations and reach new customers. As a result, its online sales have seen an average annual growth rate of 32% over the past three years [17]. Another example is Villa Lagoon Title in Gulf Shores, Alabama, which used eCommerce tools to grow its customer base in Asia, Europe, and the Middle East; today, its international shipments now comprise 15% of its total sales [17].

As we describe in the following, the literature offers many examples of how SMEs can employ eCommerce to great benefit in three key business areas: operations, sales and marketing, and finance. However, existing literature offers little guidance on how global

disruptions might impact eCommerce use in these areas. This is why our study seeks to understand the cross relationships between a pandemic and eCommerce within the context of an SME.

Appendix A summarizes the literature's findings regarding the specific effects of eCommerce on the three SME business areas. We describe these literatures in the following. We then focus on these three areas to evaluate our research questions and assess SME performance in each area before and during the global disruption.

While our contribution to literature is to examine how disruptions impact key SME business areas, our contribution to practice is to present post-pandemic recommendations to SMEs on how to alleviate the effects of a global disruption and thus help them stay afloat amidst the crisis. To frame our discussion, we describe here how existing literature categorizes eCommerce contributions to SMEs in operations, sales and marketing, and finance.

While prior publications often describe the benefits of eCommerce, some articles describe the inability of SMEs to fully capture these benefits, as well as negative effects of eCommerce use. Given these divergent findings, our discussion later offers recommendations that SMEs can use to ensure that eCommerce benefits will outweigh the potential challenges its use presents in each of the three major business areas.

Further, the literature's focus on disruptions is limited to business disruptions. In contrast, our research question aims to begin unravelling the effects on SME business areas of a larger disruption—that of the global COVID-19 pandemic.

### 2.2. Operational

#### 2.2.1. Business Efficiencies

Information technology (IT) has long been regarded as a breakthrough in improving business efficiency and effectiveness in domestic settings [18]. eCommerce is a cost-effective way to streamline business processes in smaller organizations, allowing business owners to interact directly with their partners and customers. eCommerce promotes internal efficiencies by reducing the amount of paperwork, enforcing data integrity, reducing the number of errors in data processing, and improving distribution channels [19]. eCommerce allows SMEs to use available resources more effectively [2] and make market transactions easier and more transparent [20]. It can potentially reduce transaction processes and coordination using electronic data interchange (EDI) interfaces and using disintermediation, which removes segments of the transaction process [21].

An SME's smaller structures allow for agility, creation of niche strategies, and shorter reaction times, but it also limits robustness to weather disruptions, and requires SMEs to focus too much on the existing business, rather than risk incurring operational losses on new activities [3].

#### 2.2.2. Technology

eCommerce platform technologies bring several advantages to SMEs. First, they make information flow more efficiently [22]. For example, through customer communications, establishing customer profiles, and cooperating with customers, SMEs can obtain targeted, unique data and valuable market information, which can help them with business decisions. Technology also allows SMEs to exchange and share information securely using technologies such as EDI [23] and thus makes market transactions easier and more transparent [21]. eCommerce is an easier, cheaper way of doing business because customer orders can be accepted, confirmed, processed, and paid in an online environment [19]. The benefits that information technology (IT) brings are even more critical in international business activities, because they overcome cultural and language barriers. eCommerce use also alleviates challenges due to geographic dispersion and time zone differences among participating organizations, while providing visibility to all stakeholders in a business transaction.

Findings in other literature indicate that the downside of technology for SMEs is their failure to regularly update the website due to shortage of capital and skilled personnel [24].

2.2.3. Supply Chain

eCommerce lets SMEs streamline their supply chains and activities [25], enabling supply chain activities such as inventory management, order processing and fulfillment, and communication with multiple stakeholders. These activities include production processes and the delivery of goods and services to various parts of the globe. An example solution is the advancement in order fulfillment technologies, which have helped integrate front-end business processes. On the supply side, companies can purchase material inputs and services more efficiently. With these integrated order fulfillment systems, more suppliers can access global value chains (GVCs), including businesses in geographically dispersed areas [21]. GVCs are a full range of activities with which firms and workers engage to bring a good or service from its conception to its end use and beyond. GVC activities include those related to producing, distributing, and transporting the product (supply chain), as well as "intangible" activities such as research and development, design, marketing, and support services [26].

*2.3. Sales and Marketing*

2.3.1. Marketing Strategies

As businesses continue to trade online, more and more SMEs are using eCommerce as a sales and marketing tool. SMEs use eCommerce websites as a sales channel between business, its competitors, customers, and the world in general. The main objective of eCommerce websites is to market products and services [27]. Gandour stated that websites are useful for SMEs only if they create value for the customer. A disadvantage here is that SMEs using marketplaces instead of their own platforms cannot control site content in a way that allows them to provide sufficient value to customers.

Marketing strategies reflect how SMEs position themselves and compete in the market. Such strategies are a strong indicator of an SME's responsiveness to its customers. Strategies guide managers perform activities that promote marketing goals [4]. Marketing strategies are critical for an SME to capture a larger share of customers in a highly competitive environment. eCommerce allows SMEs to compete with larger organizations, who have the scalability to reach a larger audience and use multiple channels that provide detailed information about their company and the products they market [28].

Because SMEs typically have scarce resources, it is imperative that they expand to a larger customer base at a reduced cost. eCommerce reduces the cost of advertising. It links to social media through search engine marketing (SEM), which is often used to attract more customers. While SEM will be a good channel to advertise [28], social media platforms are saturated and may also direct customers to other sites or vendors. Consumers use search engines on the internet to find products. The search results become visible through advertisements or sponsored links, or organic links. Advertisers increase their visibility by controlling algorithms for search patterns using search engine optimization, or SEO [29]. This technique is critical for SMEs to reach a wider network of customers.

Finally, eCommerce allows SMEs to gather data for data analytics. This is imperative for business growth and revenue generation. SMEs can collect a wealth of primary marketing data automatically from a good website by recording customer purchasing behavior, product selection, and payment information [30]. They can also use data analytics tools to measure consumer behavior to direct their marketing strategy [27]. Despite the availability of information through these tools, however, the lack of capital and skilled resources among SMEs still becomes a challenge. SMEs are not equipped with resources to regularly update website and product information [24].

2.3.2. Globalization and New Market Entry

Globalization creates opportunities for SMEs to grow their businesses, while eCommerce allows SMEs to penetrate remote markets without requiring a physical presence [21]. The internet and eCommerce help SMEs to compete with larger counterparts in global markets by overcoming distance and size [31]. Communication and transportation technology

have improved over the past two decades, letting organizations project brand awareness beyond their immediate market. This process made international markets more accessible to SMEs. The emergence of eCommerce has provided a wide range of retailers with a powerful marketing channel that can reach consumers world-wide [32]. Despite SME efforts to improve marketing strategies to cater to a globalized world and enter new markets, however, increasing competition with foreign enterprises may pose a risk. Globalization creates greater risks if an SME is unable to improve quality, cost competitiveness, and management practices [3] to compete in the international business arena.

### 2.3.3. Customer Service

eCommerce allows SMEs to obtain 24/7 availability to consumers despite geographical distances. This increases convenience, letting customers shop anytime from anywhere in the world [24]. Local and global competition has forced SMEs to implement eCommerce for survival and has increased competitive pressures [28]. At the same time, using eCommerce platforms in marketing efforts reduces SME dependence on agents and distributors [21], while enhancing customer engagement and communication to support internationalization strategies [33].

ECommerce platforms serve as a communication channel for bidirectional information transfer and communication [27]. Through customer feedback, an SME can improve product quality [11,31] and also respond immediately to that feedback, which strengthens customer relationships [27].

Social media platforms help create a two-way conversation that lets brands engage their customers and respond to their needs [28]. This feedback system results in good customer service that yields customer retention and establishes stickiness. Stickiness, which connotes customer or brand loyalty, is a vague measure for an organization's ability to convert visitors to customers and retain existing ones [34]. As research shows, brand loyalty has an influence on brand behavior. Establishing brand loyalty is still a challenge for SMEs since the online stores do not meet the needs of customers. Reasons include inconvenient navigation, information overload, and design issues [35].

Customer satisfaction and attention to customer behavior are critical factors in SME eCommerce [36]. The internet offers customers many options, so SMEs must maintain suitable communication and feedback times to retain customers. eCommerce platforms substantially reduce customer waiting times [21], leading to quick responses, faster resolutions, and customer loyalty. SMEs fail to reap the benefits of eCommerce to increase response time to customers. For customer communications, most SMEs prefer to respond through using emails or phone calls [24].

### 2.4. Financial

### 2.4.1. Revenue Growth and Cost Reduction

Prior to the internet, market expansion was an expensive, tedious process with minimal odds of success. Businesses had to commit significant capital to market research before even venturing out. During execution, expansion required considerable investment in operations and physical assets. The eCommerce revolution has replaced research with data analytics. Globalization now allows SMEs to expand on an asset-light basis. On the revenue side, the ability to reach more customers remotely drives up revenue without a corresponding rise in costs, leading to lower cost of revenue expansion. B2C e-commerce opens the possibility for producers to capture higher margins, again by disintermediation in the international distribution of goods.

Technical barriers such as technological literacy, however, deter SMEs ability to optimize eCommerce platforms. To alleviate these barriers, SMEs need to employ additional manpower or outsource processes for implementation, along with connection costs, hardware and software upgrades and maintenance costs [3].

### 2.4.2. Improved Working Capital Position

SMEs can gain a competitive advantage from optimizing cash flow and managing working capital, both of which can be facilitated by eCommerce. Deploying eCommerce technology can drastically minimize SMEs operating costs, or the day-to-day costs of running a business. In one of the most obvious gains, the ability to communicate in detail about demand and supply significantly reduces the need for transportation and movement of goods. More importantly, the flow of information up and downstream improves operational synchrony, reducing operational inefficiency by ensuring that more actions and decisions are based on signals from the market, the supply chain, or the value chain. This minimizes the frequency of resource wastage and leakage. In effect, operational efficiency reduces working capital.

### 2.4.3. Financial Management and Asset Monitoring

Accounting and fiscal management costs have historically kept financial management by SMEs at a minimum. Physical records were difficult to obtain and manipulate. However, the digitization of eCommerce operations has baked in digital recordkeeping in SME operations. It positions the business effectively for global financial monitoring and management, guiding decision-makers on investment decisions. eCommerce reduces the cost of business operations [2] while improving SME revenue streams. In effect, it improves the business's bottom line. Among the tangible benefits is reduced production cost [19] by leveraging available data in eCommerce platforms in financial forecasting, demand planning, resource allocation, and production management. eCommerce has been found to reduce transaction and coordination costs by utilizing online databases globally [37], allowing SMEs better financial management structures, processes and asset monitoring.

### 2.5. The Cons of Ecommerce and Global Disruptions

Literature over the past decade has presented pros and cons of eCommerce use. With the drastic effects of global disruptions such as the pandemic, we aim to determine how SMEs use eCommerce to support functions when human resources are relatively scarce, as well as to identify how global disruptions impact the key SME functions.

Of the many downsides of adopting eCommerce and digital operations, a business may become regimented to decisions framed around remote operations. A good example is in logistics. Typically, businesses using eCommerce outsource all or part of their logistics operations. For businesses that totally outsource, the day-to-day decision-making on their logistics is made by the service provider and can become obscure to the business owner. Thus, when opportunities arise to make minor operational decisions that can help the business, decision-makers are either unaware or under-equipped to take such decisions. Disruptions however provide a practical context for businesses to exploit such opportunities. A disruption, by placing the service provider in a position where they cannot act quickly with off-the-shelf solutions, creates both the problem and the need for the client business to engage directly in the solution. Supply chain disruptions caused by COVID-19 offer a great example. With logistics providers unable to move goods for a significant period, many businesses had the opportunity to reassess their supply chain activities and explore new ideas for business continuity [38].

Besides this, eCommerce can cause a reduction in the quality of information that businesses receive. This may seem paradoxical, since eCommerce on the one hand improves the ability of businesses to communicate with more customers. However, since only about 30% of communication is verbal, information collected via eCommerce can be severely regimented. This is especially significant for businesses that trade in physical goods, for whom instant feedback on the customer's first contact with their product is critical. Losing access to such information can hamper product improvement, customer satisfaction and revenue growth, thus diminishing returns on investment. [39].

The downsides of eCommerce from financial management and monitoring are limited but exist all the same. The physical nature of brick-and-mortar businesses prompts man-

agers and business owners to scale their administrative and oversight infrastructure in line with their growth. Naturally, opening more physical stores to tap into new geography carries along administrative capabilities. This is not the case for eCommerce, where growth can very quickly outpace infrastructure. It has adverse implications for financial management, monitoring and compliance. A total of 32% of failed eCommerce businesses said they failed because they ran out of funds to drive their aggressive growth [40]. For the businesses that succeed, many run suboptimal financial structures that do not take advantage of all their opportunities. Underpinning these, the majority of eCommerce businesses are so overwhelmed by the financial data available to them that they are unable to maximize their benefits. On the sum of all this, eCommerce can go from financial boom to financial burden for SMEs [41]. Here again, natural disruptions can prompt SMEs to assess their financial situations and their projected needs. The shocks generated by disruptions can serve as a natural stress test for small businesses, revealing weaknesses in their financial plans.

However, disruptions can be a double-edged sword for businesses. The COVID-19 pandemic is a natural disaster that has created many disruptions in human lives and in global businesses. Some businesses are riding the pandemic wave to increased success, while others experience a decline. The pandemic has created a domino effect of multiple disruptions—most notably in the supply chain. Whether natural or deliberate, disruptions hamper eCommerce activities to connect producers to consumers across the world. This makes the relationship between eCommerce and the pandemic worth studying.

## 3. Methodology

For this study, we used business case study research, which attempts to study the subject matter in context and use empirical evidence from one or more organizations [41]. The case study method in this research allows us to present a practical application of ANT. We describe technology use in terms of all actors in an SMEs eCommerce network, which has few human actors. Our goal is to analyze the disruption caused by the pandemic on the supply chains of SME eCommerce in the United States in relation to their major business functions. We chose to use explanatory case studies as our research method, as they provide explanations as to how and why business decisions are made and why the decision-making process works the way it does [42].

Our research is a multiple-case design that uses replication logic, wherein we conducted a series of interviews to study finding replication across cases. We used literal replication logic to test the theory among the case study participants, similar to a series of experiments to test existing theory [43]. Using this research design allows us to confirm and contrast the impact of eCommerce. Our research questions focus on *how* and *why*, to understand why business processes in each organization work the way they do [41], with each SME organization as the unit of analysis in our multiple case study design.

We followed a series of semi-structured questions but did not limit the interviewee's answers. To avoid confusion because of industry and trade jargon, we explicated terms to define the ambiguous statements to encourage elaboration. We addressed specific questions and encouraged follow-up statements to determine possible commonalities and differences among organizations. We designed questions as linear and open-ended, allowing the case study participants the opportunity to expound on their answers during the interview.

Each SME in our case study trades or sells distinct consumer goods and some services associated with the products they sold. We chose participants who engaged in the trade of different products. This criterion is a deliberate process to avoid duplication of results between two competing companies or trading similar products. It was also imperative that we validate the number of employees among the respondents to identify the effect of limited resources in the proper application of technology and the theoretical implication of ANT in a SMEs business model. We respected some of the respondents' choices to avoid certain questions because of confidentiality or uncertainty.

### 3.1. Reliability and Validity

We took several steps to ensure the reliability of our case data, the validity of our empirical concepts, and the external and internal validity of our analysis. We used interview transcripts from eight SMEs as empirical evidence to define the importance of eCommerce in these functions during the disruption in their supply chain.

Following Yin [42], we improved reliability by (1) organizing each SME's case records in the same way; (2) using multiple observers to take notes during conversations with case informants; and (3) creating a case study database. Each researcher carefully documented these procedures and followed the steps carefully. All three researchers were present at all interviews. We also reviewed the cases by studying one case after completing another interview. Using the documentation for each case, we developed a cloud-based case study database to avoid documentation issues.

Inter-rater reliability addresses the issue of how consistent the answers provided by the respondents were. Text answers to open-ended questions from the interviews as were conducted by us can be analyzed to find the presence of any inconsistency [44]. The answers to the open-ended questions received from the eight cases were separately and independently assessed by three members of the research team. Keeping the objective of the study in mind, each of us rated the range of the answers for each question with a numerical scale as detailed in the following main text Section 3.5.5.

We enhanced construct validity by using multiple sources of evidence, which we gathered from different stakeholders and their corporate websites, and by establishing a chain of evidence. The sources of evidence include websites and external trade references, news clippings and articles gathered from external sources, archival documents, and information from related organizations.

As Samaddar and Kadiyala [45] explain, "external validity refers to establishing the domain into which a study's findings can be generalized. Such validity is achieved through statistical generalization in a survey research and through analytical generalization in case research. Such analytical generalization should not be expected to be automatic and can only happen by using some replication logic and by conducting new cases studies (or experiments)." As Yin points out (p. 44):

"A common complaint about case studies is that it is difficult to generalize from one case to another. Thus, analyst falls into the trap of trying to select a 'representative' case or set of cases. Yet no set of cases, no matter how large is likely to deal satisfactorily with the complaint. The problem lies in the very notion of generalizing to other case studies."

Thus, unlike survey research, a true external validity of a set of case studies (like 8 cases in our study) can only be established by conducting other case studies to generalize, and thus, weaving an overall theory. Our set of 8 case studies is itself an experiment to (externally) validate and thus to generalize findings from the already existing literature. Its replication logic is driven by theoretical heterogeneity between contexts of our study and others reported in the literature. Any findings of our study that falsify and support previous findings, as expected in case research, are expected to stand on their own in generalizing previous findings. External validation of our findings into other organizations including SME's can only be done when the findings of our case study are known, and subsequently other firm case(s) are studied to generalize the results of our study. The replication logic then will be driven by theoretical homogeneity of ecommerce contexts of SMEs in the US and abroad.

### 3.2. Sampling

To select our sample, we used a combination of methods: representative sampling and convenience sampling. First, our research questions were about small and medium enterprises (SMEs). We first identified different attributes of SME's that will fit the bill of our research requirement.

We first focused on organizations with no more than 70 employees. We then determined which of those organizations use eCommerce to sell their products. From those firms,

we chose eight organizations that had been in business for at least two years to provide compelling support for our initial propositions. This is in keeping with Yin [42], who suggested that researchers consider 6–10 aggregate cases to pursue different replication patterns. All eight of the SMEs we selected sold consumer products, or final goods, which are products that individuals buy for personal, or households use. Given that our research question focused on the effects of supply chain disruption, we examined SMEs that are using either their own eCommerce platform or an eCommerce marketplace.

Then, we used the convenience approach to target companies that will be our subjects. Case studies do require in-depth access to the subject companies, written and spoken answers, access to documents, etc. Such in-depth access in turn requires deeper cooperation and reasonable trust from the company management/owners. Consequently, our approach was to use our own connections and familiarities with known SME top management/ownership. The eight companies studied here were the ones who agreed to participate in-depth in our study and we took that offer. Here the convenience was in the form of access and cooperation from the companies that would improve the feasibility of the study. Both representation and convenience reasons have been taken together and we found that the eight companies were reasonable and adequate samples for our research.

The eight SMEs operate in three key sectors: retail, food, and consumer goods. These organizations all use eCommerce as a tool for selling products and reaching their clients. They differ, however, in key dimensions such as industry, product category, business model, eCommerce platform, and target market. We chose the organizations to represent a diverse sample of SMEs that use eCommerce. For each firm, we interviewed at least three employees who are involved in various functions necessary to our analysis level. Table 1 offers more details about our cases.

**Table 1.** Company profiles.

| Company | Industry, Location | Profile | Key Personnel |
|---------|-------------------|---------|---------------|
| **Case 1** | Food product manufacturer, Toccoa GA | 1–5 employees, 7 years in business, private business, sales less than a million, B2C/B2B, Shopify eCommerce | 1 female owner |
| **Case 2** | Coffee Roaster, Tallahassee, FL | 10–20 employees, 11 years in business, private business, sales over a million, B2C/B2B, Shopify eCommerce | 1 male owner |
| **Case 3** | Home products, Venice, CA | 20–50 employees, 8 years in business, private business, sales less than 5 million, B2C, Shopify eCommerce | 1 female owner |
| **Case 4** | Hair styling products, Atlanta GA | 1–5 employees, 16 years in business, private business, sales less than a million, B2C, Square eCommerce | 1 female owner |
| **Case 5** | Garment products, Reno, NV | 10–20 employees, 20 years in business, private business, sales less than 5 million, B2B, third party eCommerce | 2 male owners |
| **Case 6** | Honeybee products, Toccoa GA | 10–25 employees, 10 years in business, private business, sales less than 5 million, BC2, WooCommerce eCommerce | 1 male owner |
| **Case 7** | Wood products, Toccoa GA | 20–50 employees, 40 years in business, private business, sales less than 10 million, B2C/B2B, Craft Commerce eCommerce | 2 owners (wife and husband), 1 senior executive |
| **Case 8** | Fishing products, Chicago, IL | 100–150 employees, 10 years in business, private business, sales over 300 million, B2C, Magento eCommerce | 1 male owner, 6 executives |

*3.3. Data Sources*

Primary and Secondary Data

Our data collection was based primarily on semi-structured interviews, and we followed a protocol to increase case reliability. We complemented this data with reviews of company documentation and industry-related information. We followed a two-phased interviewing process. In the first phase, we interviewed either the CEO or the business owner. When talking to the highest-ranked executive in an organization was not possible, we interviewed another member of the executive team. In most cases, this was sufficient to uncover the information needed for the research. In a few of those cases, however, we had questions that only the owner or the CEO could answer. In such cases, we conducted abridged interviews by phone or email with the business owner or CEO, which demanded less of their time and afforded us the opportunity to obtain the information we needed.

*3.4. Case Study Protocol*

We followed Yin's framework and interview protocol, which includes an overview of the case study, data collection procedures, protocol questions, and a tentative case report outline. The questions were primarily how and why questions to support a process of experimentation.

To provide the respondents with an overview of the case, we provided a letter of introduction articulating the objectives of our study, explaining that their participation may eventually aid SMEs to leverage the use of eCommerce in core business functions, particularly during a global disruption such as the COVID-19 pandemic. We also presented them with a waiver of confidentiality to guarantee the security that their name and the names of their company will be withheld from publication. We opted to coordinate with the highest-ranking person in the organization, or a delegate, to ascertain approval of participation. The letter we provided included the contact information of all research participants.

Prior to the interview proper, we searched secondary data from company websites and trade data sources to familiarize ourselves with company information. We performed this to validate the criteria based on the number of employees, the use of eCommerce in the distribution and selling of their products—either through their website or an eCommerce marketplace, products and services, distribution channels, and pertinent information that may lead to exploring more information about their business functions.

We scheduled the interview based on the availability of the main interviewee or the delegate. Understanding that the pandemic creates untoward interruptions in business operations while it was going on, we accommodated changes to the interviewee's schedule or shortened the original schedule.

We used a line of inquiry focusing on the case based on our protocol questions submitted earlier to the interviewee or the delegate. It was essential that we kept in mind a tentative outline to determine that the line of questioning is focused on the core functions of the study. For this reason, we followed a conventional linear sequence in data collection.

While the interview was going on, we also validated information we gathered from secondary sources. We also employed a cross-case analysis to compare a response by another SME to determine themes or disparities in their responses.

The lockdowns during the pandemic prohibited face-to-face interviews, giving rise to technology-related challenges such as technical difficulties, connection problems, or interruptions during the interviews. Nonetheless, we completed the interviews within the timeframe committed.

We utilized the new normal of using zoom or web-conference meeting platforms, which allowed us to acquire audio and video recordings while one of the observers takes copious notes of important information and follow-up questions if necessary.

### 3.5. The Case Study

3.5.1. Selecting the Case Study SMEs

We chose organizations from a roster of potential study participants. To screen the potential participants, we collected archival data by interviewing contacts within our own organizations, reviewing documentation such as standard operating procedures (SOPs) and information on company websites and third-party sources. We then shortlisted the candidates for the research based on the following relevant criteria:

- Has 70 employees or fewer
- Uses eCommerce platform for one or more functions
- Employs different parties to handle various functions in the organization

Because some of the SMEs are direct customers of our employers, we presented our intent to include certain customers who are willing to participate in the study. We presented our employers a third-party waiver of confidentiality and a statement that our research is for doctoral studies. We also advised the case study participants that our employers were not involved with this research. Through this process, we chose the eight SMEs that participated in our case study.

3.5.2. Data Collection Procedures

Before requesting interviews, we requested introductions from our contacts at each of the organizations and presented an introductory letter. We sent letters to 24 SMEs, from which we eventually selected 8. For companies with whom we did not have a direct relationship, we engaged colleagues from our organizations to provide referrals. Our contacts continued to manage the relationship between the SMEs and one of our three-member research team throughout the process. This approach also allowed us to perform a screening process for case study interview candidates.

We collected data from SME employees on their everyday situations using semi-structured interviews. We first sent an email communication to the interviewees in preparation for the interview proper. This allowed the interviewees to gather responses from other stakeholders within their organization. We then scheduled interviews with each person so that we could focus on their individual responses—again testing the reliability of the case.

We conducted initial virtual interviews, for a period of three months from 11 August 2021, until 5 October 2021. The interviews lasted for an hour to two hours for each case. We advised the interviewees that a possible follow-up might be necessary for clarifications or additional information. We identified gaps during the data analysis and interpretation stage requiring us to conduct follow-up virtual interviews with 4 of the SMEs during December and January 2021. The follow-up interviews lasted for half an hour.

In addition to actual questions, we asked our interviewees to share additional documents that might provide more details about their responses. We used these documents—either a hard copy or an electronic version—as part of our explicit data collection approach to corroborate and augment information gathered during the interviews [42].

3.5.3. Protocol Questions

As Appendix B shows, our interview questions started with general questions about the organization to establish the knowledge and credibility of the interviewee. We used the protocol questions as a line of inquiry, not as a survey questionnaire. This allowed the interviewees to expand on their responses in their own words. We conducted follow-up interviews, which are abridged versions of the original interview to validate responses and deliver cross-case validations based on initial responses, between similar-sized organizations or interviewees with similar responses.

To establish a process model, the questions focused on describing specific focus areas that may have been impacted using eCommerce in three different periods: (1) before the pandemic; (2) during the pandemic; and (3) in terms of plans or projections for when the pandemic is over. We provided specific timelines such that "before" was prior to March 2020 and "after" will be when their business operates in normal conditions.

### 3.5.4. Data Processing and Analysis

We recorded and transcribed all interviews, then processed the transcripts in two ways: through a manual analysis, and through an automated analysis. For the manual analysis, we recorded each interviewee's response next to the target question in an Excel spreadsheet. Each researcher separately interpreted the answers of each of the SME responses. The interpretation determined inter-rater validity on whether the response supported literature or rejected it.

Once this processing was completed, each of us interpreted the responses of the interviewees for each question, briefly summarizing whether the response supported, rejected, or was neutral about the target question. We also interpreted each response based on the research question on the impact of eCommerce on SMEs supply chain disruption identifying its benefit to operations, sales and marketing, and finance.

### 3.5.5. Scoring

The next step was to interpret and analyze the responses. By design and in line with our research objectives, our interviews produced qualitative responses. In reaching conclusions about each question in the questionnaire, we adopted a scoring system as follows, based the conversion method developed by Franceschini and Galetto [46] for converting qualitative data into ordinal quantitative data.

To this end, we adopted a quantitative scoring system for rating the responses. To achieve inter-rater reliability, we used multiple raters and then achieved an aggregated score of their ratings across each question for 8 cases. To avoid any fence-sitting tendency—a state of indecision or neutrality with respect to conflicting positions or neutral score—we used an odd number (three out four) of the researchers to serve as the raters. We conducted the scoring and aggregation in a four-step process, progressively focusing on the following levels of data. Table 2 offers more details about our scoring process.

**Table 2.** Steps and Levels of Data.

| Step | Description | Output |
|------|-------------|--------|
| Step 1 | Each Individual Response is scored −1, 0 or 1, by each rater | These are the rater scores |
| Step 2 | Results from step 1 are added up for each response to arrive at 8 scores per Question for each of 8 businesses interviewed | These are the response scores |
| Step 3 | For each Question, all the 8 scores in step 2 are averaged to obtain an aggregate score | These are the question scores. |
| Step 4 | To capture the band of inconclusive results, a buffer zone of 10% of the extreme scores | This yields a band of −0.3 to +0.3 around zero. Conclusions are drawn based on this band as described in the discussions |

Step 1: Based on the interpretation, each of the three authors, serving in this case as raters, scored each response by each interviewee on a three-point scale. We used a positive-zero-negative scoring scale to establish directionality of agreement and magnitude of validity. Thus, each question could be scored −1, 0, or 1 by each author. We chose this scale because we found that using one index (a positive number 1, 2, or 3) to track two attributes was impossible. The value of the score shows validity, while the sign (positive or negative) showed the direction of agreement between raters. A neutral score was interpreted as indifferent or inconclusive. Table 3 provides an example of the first step in the scoring process.

**Table 3.** Step 1.

| Question/Rater | 4. Tell Us about Your Products- What Are the Top 3 Products | 5. What eCommerce Platform Do You Use | 21a Is Your Website Utilized for SEO If Not, Why Was This Utilized before the Pandemic |
|---|---|---|---|
| **Case 1** | | | |
| **Rater 1 Score** | 1 | 1 | −1 |
| **Rater 2 Score** | 1 | 0 | −1 |
| **Rater 3 Score** | 1 | 1 | 0 |

For Case 1's answer to question 4, Rater 1 rates the answer be valid, hence +1, Rater 2 rates the answer be valid, hence +1, Rater 3 rates the answer be valid, hence +1. For Case 1's answer to question 5, Rater 1 rates the answer be valid, hence +1, Rater 2 rates the answer be inconclusive, hence 0, Rater 3 rates the answer be valid, hence +1. For Case 1's answer to question 21, Rater 1 rates the answer be invalid, hence −1, Rater 2 rates the answer be invalid, hence −1, Rater 3 rates the answer be inconclusive, hence 0.

Step 2: Next, we computed an aggregate score for every question in every case, by adding the scores from each author. Table 4 shows step two and the aggregate score example.

**Table 4.** Step 2.

| Question/Rater | 4. Tell Us about Your Products- What Are the Top 3 Products | 5. What eCommerce Platform Do you Use | 21a Is Your Website Utilized for SEO If Not, Why Was This Utilized before the Pandemic |
|---|---|---|---|
| **Case 1** | | | |
| **Rater 1 Score** | 1 | 1 | −1 |
| **Rater 2 Score** | 1 | 0 | −1 |
| **Rater 3 Score** | 1 | 1 | 0 |
| **Aggregate score** | 3 | 2 | −2 |

Example: For Case 1's answer to question 4, the aggregate score will be (+1) + (+1) + (+1) = 3. For Case 1's answer to question 5, the aggregate score will be (+1) + (0) + (+1) = 2. For Case 1's answer to question 21a, the aggregate score will be (−1) + (−1) + (0) = −2.

Step 3: We then averaged all aggregate scores for all cases for each question to get an average score per question. Finally, we counted the number of cases that supported and rejected each question, respectively. Table 5 shows step three and the classification example.

Example: Question 4 is supported by the aggregate score of 4 cases, and across all 8 cases scores an average of 1.25. Question 5 is supported by the aggregate score of 8 cases, and across all 8 cases scores an average of 2.63. Question 21a is supported by the aggregate score of 6 cases, and across all 8 cases scores there is an average of 2.00.

Step 4: For the average aggregate scoring, we observed that no aggregated figure was zero despite the variations in scoring—which implies that no question was inconclusive. This is problematic because a slightly positive or slightly negative average aggregate can misrepresent the answer. For example, if we have 7 zero scores and 1 positive score, we cannot conclude that the score was positive; the same applies to negative scores. Additionally, questions with strongly polar scores (such as +3 and −3) that are evenly distributed across eight responses may not fall exactly on zero, even though they should be inconclusive.

Because there were only a few of these situations, we introduced a margin of error to make our conclusions more accurate. We chose 10% because it does not eliminate the contribution of each such answer (about 12% of the eight responses) and thus established a margin of error on either side of zero (−0.3 and +0.3). In this way, we defined our region

of uncertainty or inconclusiveness: only scores above +0.3 are supported, and only scores below –0.3 are rejected. Scores between –0.3 and +0.3 are inconclusive.

**Table 5.** Step 3.

| Question/Case | 4. Tell Us about Your Products- What Are the Top 3 Products | 5. What eCommerce Platform do You Use | 21a Is Your Website Utilized for SEO If Not, Why Was This Utilized before the Pandemic |
|---|---|---|---|
| **Case 1** | 3 | 2 | −2 |
| **Case 2** | 1 | 3 | 3 |
| **Case 3** | 3 | 3 | 3 |
| **Case 4** | 0 | 3 | 3 |
| **Case 5** | 0 | 3 | 0 |
| **Case 6** | 3 | 2 | 3 |
| **Case 7** | 0 | 2 | 3 |
| **Case 8** | 0 | 3 | 3 |
| **Aggregate Average** | 1.25 | 2.63 | 2.00 |
| **Supports** | 4 | 8 | 6 |
| **Rejects** | 0 | 0 | 1 |

Table 6 shows how we scored the interview questions, while the tables in Appendix C offer extensive details about our scoring of questions, including the progression of scoring on questions 1 and 2.

**Table 6.** Scoring responses to the interview questions.

| Question | Score | Meaning |
|---|---|---|
| Single Question | 1 | Question is answered and supported in the affirmative. Affirmation is in the direction of the overall research question on positive contribution of eCommerce to SMEs. |
| | 0 | No basis in the response for answering question affirmatively, or the answer is not relevant to the question at all. No definitive conclusion on the validity of the affirmative answer. |
| | −1 | Question is answered in the negative or an alternative explanation is given for observed trend. |
| Aggregate (Sum Across 3 Authors) | 3 | Perfect alignment on affirmative answer; supports literature |
| | $0 < X < 3$ | Affirmative, albeit imperfect |
| | 0 | Neutral or inconclusive |
| | $-3 < X < 0$ | Negative, albeit imperfect |
| | −3 | Perfect alignment on negative answer; rejects literature |
| Average Aggregate (Average Across 8 Cases) | $0.3 < X < 3$ | Supported |
| | $0 < X < 0.3$ | Inconclusive (within +10% margin of error) |
| | 0 | Neutral or inconclusive |
| | $-0.3 < X < 0$ | Inconclusive (within −10% margin of error) |
| | −0.3 to −3 | Rejected |

For a comprehensive illustration of the scoring process, see also Appendix D.

## 4. Results

With our objective being to evaluate the impact of eCommerce on three major functions in an SME organization, particularly amid a global disruption—the pandemic of 2020—we unraveled two related findings that validated the research question.

First, eCommerce effectively supplemented the shortage of human employees in smaller organizations. The eCommerce platform served as a non-human element in the network and enabled effective processes, with human actors in the organization adjusting and reacting fluidly based on the processes that the platform enables. Second, the effectivity in using eCommerce for SMEs was intensified during the recent disruptions. Our case study findings exemplify its importance in that the SMEs we studied plan to expand the use of eCommerce in fragmented functions during the pandemic.

### 4.1. Operational

Our findings generally—though not universally—show support for eCommerce having a positive impact on various aspects of SME operations. Most of the case studies have not realized the potential contribution of integrating systems, tools, and processes in operational functions that have been presented in previous studies. Some of our cases recognized the advantage of using their eCommerce platform during the pandemic, when human resources are scarce, and thus considered integrating these processes into their eCommerce platform.

#### 4.1.1. Processes

eCommerce is known to be a cost-effective way to help organizations streamline their processes through effective use of its limited resources [2]. However, not all SMEs in the study invested in integrating operational business functions into their eCommerce platform. Data are captured from platforms to support processes in drop shipping, identifying inventory levels, and decision making. Prior research has shown that eCommerce platforms reduce paperwork, enforce data integrity, and reduce the number of errors overall. One of the case study participants realized the need to integrate fragmented systems, stating that "*[if] all [data is] in one place, [it] makes it more effective.*" Another stated that "*it helps you know your inventory levels, get data on sales coming in [and on the] best sales items for driving decisions.*"

Two of the eight cases experienced no improvement in their business efficiencies in using eCommerce platforms when the pandemic hit. The SMEs continued to operate using various systems in among the three key functions. One case study participant changed its business model completely during the pandemic from brick-and-mortar stores to online sales during the lockdowns in order to stay afloat. The business owner stated, "*We went from not having anything online [to having] it during pandemic.*" Another SME made a strategic change of eCommerce platforms to enhance its efficiencies.

When the pandemic ends, three of the eight cases are planning to make changes to their eCommerce platforms to increase their business efficiencies, and one will enhance features that are not currently activated in their eCommerce platform and eventually streamline their processes.

The findings in the case study support the research question that eCommerce does help improve business efficiencies. The manual collection, collation, interpretation, and analysis of data from disintegrated sources are simplified on electronic platforms. This helps SMEs use available resources more effectively. SMEs can achieve this through EDI interchange. eCommerce gives small businesses a way to use their resources more effectively, by making transactions easier and more transparent.

Furthermore, it allows SMEs to collect a wealth of information for better decision making. In our study, five of the eight cases supported the use of available resources. As the following quotes exemplify, several use their eCommerce platform to collect information about how customers are using it, thus encouraging data-driven decision making, while another found evidence that eCommerce was distributing work more effectively but was also forecasting labor demands. As one SME stated, "*eCommerce is the engine of the business, it has to have optimum performance, and serve as [a means for] data collecting for driving decisions.*" Another said, "*Our eCommerce primarily represents 55% of the business, so is a major contribution in terms of distribution of work and tasks to be done, and how much hiring needs to be done.*"

The three remaining cases did not find an impact in their usage of available resources through eCommerce. All three cases had disintegrated business functions, which was a major challenge for each organization. One case had different information silos, which turned out to be a complex challenge; in another case, sales could be enhanced if the firm integrated the fragmented business functions, "*Well, we need to find a solution for a more centralized option for accounting/financing and the current software is not the solution, [we] had a not good experience with it. [The] Main challenge is different silos of information.*"

eCommerce also fosters an environment that allows business owners to improve or streamline the transaction process by removing unneeded internal or external steps or automating those steps using digital tools, thus allowing disintermediation, or removing a segment of the transaction process [21]. In our study, despite all the benefits we presented in prior studies, two of the eight cases rejected our argument, stating that "*no middlemen were eliminated in any of our processes*" and that they experienced "*no casualties*" before the pandemic either. During the pandemic, one case evidenced disintermediation when it had to remove brick-and-mortar retailers and switch to online sales. The rest of the SMEs realized the effect of disintermediation in using an eCommerce platform, even at a miniscule impact to their businesses. It could be as minimal as printing of shipping labels and tracking of orders to cutting down the number of brick-and-mortar stores, or as a printing company puts it "*stripping our wholesalers.*"

### 4.1.2. Technology

As the following discussion shows, we examined two issues related to technology. First, we studied the effectiveness of eCommerce in information exchange. eCommerce facilitates information exchange between stakeholders, whether they are customers, suppliers, or employees. It also provides an opportunity for companies to showcase products and provide detailed information [23]. The other issue that eCommerce should facilitate for an organization is transparency and efficient information flow [41].

In our study, six of the eight cases used digital communication between stakeholders before and during the pandemic. A distributor of coffee and beans stated, "*We do all our communication with our customers, they get all of their order confirmations through our website, they get their tracking email.*" Technology has proven that it provides transparency among all stakeholders; as one participant noted, "*As far the employees and customers, when we ship orders, it (courier company) automatically sends them their tracking number, as well as uploads into our system [so] our employees get an alert as well.*"

We also asked our participants about how they interact with suppliers, customers, and employees, as well as how they communicate with customers. One case study participant provided a general description of the information flow among its stakeholders—3PL, the eCommerce platform, customers, and employees—through various systems. The supply chain manager stated, "*Once we get tracking information, which is integrated with our logistic provider, for example, once EDI II transmissions are submitted, like saying this order got shipped today, here's the tracking information that is populated through a shopping cart platform and uploaded into automated emails that will go out to the customers. Additionally, we have our own data reporting system, which will also pull that information from our shopping cart and warehouse management systems.*"

We found that the benefits of using technology for information flow and transparency was unimproved during a disruption; two cases did not support our argument, and one said their communication with customers declined.

As authors Wilsona and Abel posited, it is a challenge for an SME to fully integrate backend and front-end systems. Our study shows that SMEs do not have the expertise and sophistication of larger competitors [47]. ECommerce offers the benefit of exchanging information more precisely and effectively using embedded technologies such as email, EDI, and other media, exchanging reliable information among entities during a transaction process is essential. Six of the eight cases have a multi-process digital flow that goes from customer ordering to shipping to customers reviewing their experience. One interviewee

relayed the process as: " ... *go to our website, pick products, add to the shopping cart, and quick checkout, then get the order within two hours, order fill [or delayed} shipped, and finally a customer review.*" eCommerce is widely used in downstream supply chain processes used for information exchange with customers, similar to another response: "*Customer places an order (customer service, online), then calls our ERP (backend of our eCommerce), then, project on the order availability in real time. If there is no stock, CSR will call the client, everything goes to allocations to an order, then allocate squares (lumber) to manufacture item, and then order to manufacturing or shipping [and a] picture [is] integrated into the eCommerce to the customer, client inspection, and then scan documentation, print label, customer received, customer review it [through a customer survey].*"

Data transparency and visibility is needed for optimum business performance based on the responses of five of the eight cases that gained visibility of their data using digital tools before and during the pandemic. Upstream processes, however, are not fully integrated with their platforms. In addition, we found no data supporting the improvement of the quality and speed of information flow using eCommerce.

### 4.1.3. Supply Chain

To test the effectiveness of how using eCommerce impacted SMEs in terms of streamlining operational functions, we studied its usefulness in supply chain processes and simplifying international transactions. eCommerce businesses synchronize their operations with different supply chains, whether these are local or international, or fully or partially integrated. It allows SMEs to streamline supply chain processes and activities [25]. Supply chains are shortened by the use of platforms and the integrating systems used by intermediaries to produce, distribute, and transport products [26]. In our study, six of the eight cases had linked their supply chain to their eCommerce before the pandemic. This linkage took different forms, from third-party marketplaces to downstream logistics processes and fulfillment centers. As one of the two micro-SMEs confirmed, "*So, we have a fulfillment center that we work with, so once they [customers] purchase, everything comes into the fulfillment center. And they do all the shipping out for us.*"

During the pandemic, small businesses—like their larger competitors—experienced disruptions in their supply chain. In our study, four of the eight cases witnessed changes primarily in shipping delays and inventory shortages and communication issues. One case offered a detailed report of the challenges:

> "*Quite a substantial amount of change there, we were incurring a lot more delays, again, from importers, and stuff like that, and port congestion. That is still a residual effect that is happening today. And it's even gotten worse since probably June, July of last year. So, we have, we're seeing more delays. I think when it first happened, we were seeing about a two-week delay, but now we have gotten to maybe a month-long delay of products coming in. So, we've had to really restructure the way we think about our production planning, and our launches as well for eCommerce products, and figure out the best way and best suitable times that we can feel confident in alerting our customers, whether it's like, delaying a video production for YouTube or, you know, sending out some—trying to pull other products that we've launched before and try to get that back out there just as a substitute.*"

Three of the eight cases plan to make changes to their supply chain when the situation normalizes. For example, one case will "*carry out more inventory to fulfill orders if an order is held up somewhere along the way.*"

Another benefit inherent in eCommerce is that it also operationally lowers barriers and simplifies operational issues of conducting business overseas. It enables SMEs to overcome business challenges related to a lack of dedicated technology and/or international knowledge of trade [48]. With the speed of the digital economy, technologies available, and global consumer approach, eCommerce is critical for reaching new frontiers. The SMEs in this study have minimal international business, which negates the arguments in prior literature. Only one of the eight participants have established a market as far as

Australia. Three others have ventured into Canada since regulations are quite similar to the United States. Of those who were able to expand into the international community, the percentage of this business segment is at most 5%. The supply chain process to distribute internationally includes challenges in exportation and regulations required, which is not fully supported by eCommerce.

### 4.2. Sales and Marketing
### 4.2.1. Marketing Strategies

Overall, the research strongly suggests that eCommerce offered advantages to SME sales and marketing activities even before the pandemic, and these advantages were strengthened when customers could not physically reach brick-and-mortar stores. For example, our case study identified benefits from creating marketing strategies focusing on digital ads, which reduce costs compared to traditional advertising. Social media also helped boost advertising and reach a broader audience, while expanding the platform's use to gather marketing data for analysis. SMEs also leveraged eCommerce to enter new markets and expand their global footprint. Literature shows that while eCommerce can help SMEs easily advertise their products, expanding internationally entails constraints such as trade regulations; to avoid risks, SMEs also need knowledge about how to trade in a new geography, which requires additional resources. SMEs in our study were able to cut the costs of hiring intermediaries to boost sales by engaging retailers, direct salespeople, and agents to distribute their products while expanding their marketing footprint.

The best use of eCommerce in SME sales and marketing activities is in customer service. Our case study analysis shows that eCommerce platforms and social media improved customer communication. The findings do not offer a strong correlation between customer alignment and enhancing product quality. However, eCommerce supported most SMEs in achieving greater customer retention and brand loyalty.

Our case studies strongly support the literature suggesting that eCommerce has a strong impact on sales and marketing activities for SMEs. Online trade has been effectively proven to generate revenue for most businesses, regardless of their product or service. Using eCommerce platforms has proven even more effective during the recent pandemic disruption and the chain reaction of disruptions that it caused to the SMEs' supply chain processes. Indeed, the businesses that were part of our research were kept afloat by eCommerce.

Marketing strategies help SMEs position themselves in the market. The literature posits the benefit of eCommerce in guiding an organization's marketing activities [4]. Inter-rater validity showed that all researchers are 100% in agreement that eCommerce supports the creation of SME marketing strategies. In our case study, all eight SMEs used digital marketing activities through their eCommerce platform before the pandemic as the primary driver of their marketing strategy.

One of the micro-SME business owners who produces and distributes probiotic products stated that, before eCommerce, *"mostly, it was just people who have already taken it telling their friends, families, and neighbors, so it has been word of mouth."* After the pandemic, the SME plans to enhance the SMEs digital marketing strategies by including training and education about the company's product through their eCommerce website. The other micro-SME is a retailer of hair care products for African American women; they said that eCommerce played a significant role in that the platform provider allowed her company to venture into email marketing directly from customer sales and customer reviews. In another case study, the owner had to make changes due to regulations on privacy data and congestion in the digital ad space, which became a challenge for the firm's marketing strategies. That SME also plans to change digital marketing strategies once the pandemic is over, including to return to ad work traffic, double its advertising budget, and implement new tools such as videos into the company's website while leveraging algorithms to expand future marketing strategies.

While the pandemic kept people at home and reliant on their computers, most of the SMEs in our study kept the same digital strategy they had implemented before the pandemic. Five (62.5%) of the SMEs did not change their digital marketing strategies and have no plans to change them once the pandemic is over.

Digital ads are the most common driver of eCommerce marketing strategies. SMEs typically have fewer resources than their larger competitors, which requires them to be creative in how they market their products. To compete with larger companies that have highly sophisticated advertising and sales channels, SMEs utilize eCommerce to advertise their products and leverage SEO, a search pattern technique that uses algorithms to increase visibility and promote products [29]. Of the eight SMEs, four or 50% focused on digital advertising and found it valuable in their marketing activities. Two SMEs opted not to focus on digital advertising and SEO; one SME claimed that digital marketing and investing in SEO is too expensive, while the other did not anticipate a benefit in using SEO for their business. The remaining two SMEs were neutral, as they are both new to the digital advertising world and provided inconclusive responses during the interviews.

Of the four SMEs that focused on digital advertising, the CEO/owner of an eCommerce printing company incurred increased advertising costs when the company hired an outsourced IT company to create metatags, campaigns, and algorithms to follow the SEO trend. This SME found it more effective to outsource rather than having their internal IT department support all of the digital marketing activities, stating that "*We have the webmaster, which is also our IT company, do the optimization, Google for business and all the, the, those types of things. So, we outsource that work, anything in regard to finding us, I outsource that*". Of the eight SMEs, two made no changes to their strategies during the pandemic, other than adding to their marketing spend to increase SEO utilization. The smaller SME prefer not to follow the trend because "*it is too expensive and not [having] enough experience,*" as one of the business owners stated.

Social media is another essential digital marketing channel that supports SME efforts to advertise their products and services. The inter-rater validity suggests a strong aggregate average for social media playing a critical role in this area. The most-used social media platform among our cases was Facebook. The SMEs also used other social media platforms, including Instagram, Snapchat, Pinterest, and LinkedIn. The SMEs said that disruptions aside, social media is always an effective channel for marketing their products. They said they are continuing to leverage social media platforms during the pandemic, and all intend to continue using social media for marketing when the pandemic is over. Three SMEs plan to improve their use of social media to market their products at that time.

Using digital avenues for product marketing also allows SMEs to gather customized content and metrics. Six out of the eight SMEs said that using analytics and data management before and during the pandemic helped attract the right customers, and that they used these tools to identify a shift in demographics. Using Google analytics, a distributor and retailer of bee-keeping equipment recognized that during the pandemic, the average age of their customers shifted from ages 44–55 to ages 35–45. The SME attributed this shift to the boredom experienced when lockdowns required people to stay at home. All six respondents who realized the benefits of data analytics said that they will keep using it as an avenue for continuous learning about their customers.

### 4.2.2. Globalization and New Market Entry

eCommerce dissolves limitations that SMEs face—due to a lack of resources—when they physically expand to new markets, including remote areas [21] and global customers [32]. Our research suggests that all eight cases were able to expand their market domestically through eCommerce. Our SMEs expressed some reluctance to penetrate newer markets during the pandemic, but 50% of them envision such market expansion after the pandemic. The expansion of the sale of coffee beans and merchandise for one of the SMEs came via the availability of their products through eCommerce. As the company's purchasing manager reported, "*We've also had some growth in our wholesale side of the business,*

*we've sold two small coffee shops along the beach here in Florida, to owners who have moved to North Dakota, or Tennessee, and they still get their coffee from us because they like the coffee. And so, they're just random little coffee shops throughout the Midwest that carry [our] coffee from Tallahassee, Florida."*

Despite the ability of our SMEs to compete with their larger competitors in the international market, those that expanded beyond the United States expanded only to Canada and Australia due to supply chain restrictions. The unfamiliarity with global regulations hampers their ability to expand to the international arena. Further, the ability to globally expand depends on an SME's products. Three companies did not explore international markets because their products involve food, which has highly regulated international compliance and transportation requirements. One SME who sells wood products online stated that eCommerce *"allows us to cultivate customers anywhere and that gives us the advantage to compete globally."* He also pointed out another advantage: *"Our competitors also exist in eCommerce, makes it more easily to evaluate and keep up with the market and competitor landscape."*

Meeting such global regulations requires resources, which still limit many SMEs from penetrating into different parts of the world. This lack of resources to explore beyond the United States is a disadvantage. However, some SMEs had products—such as linen and housewares and wood products—that faced less stringent regulatory requirements and thus were able to expand beyond their national borders through eCommerce. These same SMEs are expecting further expansion after the pandemic. The VP for marketing for one case stated that, *"the logical path would be, you know, UK, EU, Australia, Singapore, Hong Kong, it's the same path—the path for most American brands."* The other six SMEs that have not ventured out internationally have no plans to expand to global markets. Our findings suggest that despite the ability to sell internationally, the hindrance lies in physically moving the products to these locations and the challenges it brings in terms of documentation.

### 4.2.3. Customer Service

The inter-rater validity shows a strong alignment in each researcher's perspective that supports the literature. Because the internet allows 24/7 connectivity to markets, it lets businesses communicate with their customers beyond the boundaries of time and distance [24]. SMEs use eCommerce to improve customer service despite limited resources to compete with larger counterparts. eCommerce has allowed the SMEs to manage customer service functions both before and during the pandemic. Based on this improvement in customer service, respondents showed no interest in making any changes after the pandemic.

To reduce the cost of hiring customer service agents or outsourcing labor to support customer service activities overseas for lower prices, SMEs use their eCommerce platform to allow customers to request information about their products. Whether the SME uses their website, social media, and other marketplaces to reach customers, the role of eCommerce in customer service supports many activities—from responding to questions and complaints to providing additional information or updating customers on their order status. Websites and eCommerce platforms have created a shift in the dynamics of customer engagement. Where SMEs previously used distributors, retailers, and salespeople, they now rely on digital tools for customer communication. One SME said they experienced more communication from customers during the pandemic:

*"I feel like we've gotten more questions, more requests. I don't know if that's a symptom of people at home having more time to think about these things, but definitely more communication with customers during a pandemic."*

None of the SMEs plan to make any changes in customer service post-pandemic. According to the respondents, the existing communication processes with their customers are working. Out of the eight SMEs, two said they find it necessary to keep their customer service teams to handle chat, email, and calls; to manage customer feedback; and to analyze that feedback to improve their services. The SME that provides online printing services does not have a customer service team yet, so the owner and co-owner handle all email responses.

Additionally, our study shows that, despite the availability of feedback systems on websites, some customers still prefer to communicate by phone or email.

Although the literature states otherwise, none of the SMEs in our study claim to use the customer feedback they receive through their eCommerce platforms to improve product quality. The literature finds, however, that the immediate feedback that eCommerce enables leads to customer satisfaction and reduced wait times [21], which are critical factors in SMEs business [36] and drive customer loyalty. In our study, only case 7 initiated a loyalty program for frequent buyers to automatically receive a 10% discount, while case 8 provides a subscription program aimed at creating customer loyalty. The latter program automatically replenishes orders based on a frequency selected by the customers upon signing for membership. Case 6 identified an additional benefit of eCommerce that improved customer service by providing a personalized approach to customer outreach beyond responding to complaints, feedback, or inquiry:

> *"And one thing that we do that no one else does is call the customer the next day after they've placed the order and thank them for their business. And then also [we ask] if they have any questions that we can answer right then, so that there's clarity of when it will ship out."*

### *4.3. Financial*

The impact of the pandemic on the financial contributions of eCommerce to SMEs is still subject to interpretations. Amongst the authors, the average inter-rater validity of +0.89 suggests that, overall, the impact of eCommerce on SME finances is still not clear, even if it appears positive (inter-rater validity represents the average level of agreement between authors on the validity of the questions asked). Case answers show a similar divided trend, with an average 3.09 of the eight cases providing support for the questions asked, while 0.70 rejected them. This suggests that, in more than half of cases (4.21), respondents were neutral on eCommerce's financial impact on SMEs.

### 4.3.1. Revenue Growth and Cost Reduction

In perhaps the most notable trend—which contradicted initial expectations—the SMEs saw their marketing costs climb with the onset of the pandemic. Many still expect these costs to rise and stay high in the long term. This expectation reflects that the higher costs are positive for these businesses, many of whom found improved returns on their marketing spend. Indeed, the pandemic has encouraged SMEs to invest more in their marketing to better take advantage of eCommerce.

This result stems largely from the increased adoption of eCommerce by SMEs, many of whom had no digital finance processes to begin with. The higher returns on digital marketing investment boosted the overall returns on marketing investment. At least five of the businesses interviewed plan to actively increase their marketing spend in future.

In the most remarkable case, a business with strong brick-and-mortar operations and strong word-of-mouth marketing reported:

> *" . . . we've reduced the digital ads because of how solid our true organic growth traffic is coming through Google. Right. And that is offset. It is the expenses that we were spending on pre-digital ads on Facebook. Oh, okay. So, it was a net savings of $8000. That went straight to the bottom line."*

Therefore, the SME effectively increased its marketing returns while slashing its budget. The following SME comment further supports the conclusion that marketing costs are seeing much higher results:

> *"We're still up a million dollars over the prior year. Oh, there you go. So yes, there is a time and place for digital marketing. What we're seeing is, is with us being placed one or two position on Google that far outweighs any kind of advertising that you'd really need to do and still, you know, you can't grow at 120% year over year over year because you*

*can't keep up with a giant like that. Correct. I'm finding even at 60 to 70%, I'm getting worn out now seven years in a row. Seeing that kind of growth."*

In support of this outlook, all eight businesses interviewed saw an increase in their revenues across their businesses before the pandemic, especially on their digital platforms. Half of the businesses recorded revenue growth during the pandemic, which they expect to continue after the pandemic. Most notably, none of the businesses reported declines in revenues. However, trends in revenues on eCommerce channels were mixed, with a few of the SMEs strengthening their brick-and-mortar operations, even as others increased their digital presence.

The reported revenue growth has not come exclusively from increased market access. Of the eight SMEs, five found it easier to deploy pricing strategies on their eCommerce platform than in physical retail. Three businesses applied these strategies rigorously during the pandemic, and five of the eight businesses expect the trend to continue. In addition to offering businesses the ability to charge more for their products, pricing may also have driven the reported growth in customer base.

Finally, transaction costs have helped SMEs reduce friction for new and existing customers. Three of the SMEs in our study have been able to reduce their transaction costs through eCommerce. While this is not a majority, only one of the eight SMEs saw their transaction costs rise as a result of eCommerce. This SME's business was cash-based, and thus it had previously incurred no tangible transaction costs, though it had suffered more friction from indirect costs.

### 4.3.2. Improved Working Capital Position

The pandemic has been a natural stress test for the working capital position of SMEs. For some of the SMEs in our study, working capital efficiency proved to be a critical factor in eCommerce adoption, with at least two reporting an overall positive experience. Inter-rater validity was quite low, however; the three authors varied widely in our conclusions about working capital gains. The majority of the gains came from adopting new digital tools for working capital management. This is another positive impact of the pandemic on eCommerce use by SMEs.

Even for businesses with exclusively physical operations and no online customers, using digital financial management tools drives efficiency in working capital. With eCommerce-enabled businesses, however, the effects multiply. In addition to having more visibility into how working capital moves through the cash cycle, the synchronicity of expenses and earnings on an electronic platform simplifies the cash cycle and minimizes the need for petty cash. This is why these tools are rising in popularity. Of our eight SMEs, three adopted these tools during the pandemic and three more plans to adopt them in the near future.

One SME was transitioning rapidly from brick-and-mortar to an omni-channel model and noted, *"Yeah, we're hoping by August of next year, that financial software will be completely implemented . . . We've been working on it for about 18 months and, basically, eight months out of the year, I cannot work on it. But those that we are in our off-season, it's all hands-on deck, and they've pretty much given me assurances that they'll be ready to go August of 2022."*

Only one business is certain that they will not be adopting digital financial management tools after the pandemic. The implication of this is that as a result of the pandemic, SMEs are increasingly aware of the potential gains from using eCommerce in financial management. Two SMEs reported reductions in their working capital costs before the pandemic, and two expect to have recorded more reductions by the end of the pandemic. In contrast, however, two businesses saw their working capital requirements rise, and another expects a rise by the end of the pandemic. This partly could be a result of business expansion through revenue growth, which naturally raises the cost base for each business.

### 4.3.3. Financial Management and Asset Monitoring

The pandemic has forced SMEs to increase their adoption of eCommerce, with secondary effects of increasing visibility of their business. Behind the looming image of revenue gains and cost reductions, that have been the focus of the impact of COVID-19 on SMEs, increased visibility into financial operations offers a longer-term benefit to SMEs using eCommerce. In essence, an eCommerce business runs on digital operations, with records showing relationships between transactions, business units, and stakeholders. As such, eCommerce operations present a skeletal view of a business at little or no extra cost. This has proven valuable through the uncertainty of the pandemic. According to one SME, "*Before the pandemic, it was retail stores and there was not eCommerce, so it was all P.O.S, invoices, and credit cards over the phone.*"

This afforded SMEs low visibility of their business. On the other hand, another SME reported that "*We do [cash transactions] on our retail side to a certain degree, but I would say it is well under 2% of our of our business . . .* ". Another SME noted "*We can do a consolidation at month end with our CPA and then that gives us the fine details of where our spending was for that given month. And, you know, it's gone through quite intensively, just to make sure that we're not missing anything. To help move to more profitability.*"

As a result of the pandemic, SMEs increasingly recognize the value of using their eCommerce platform to track business activities. Six of the eight we interviewed declared that they now do so. Further, another business intends to implement this tracking in the near future. Four businesses found it easier to track their finances as a result. Having used the pandemic as a case study for the opportunity cost of not having such systems, SMEs appear to now be pushing more aggressively for electronic transactions. Of the four businesses that did not already use electronic transactions, two have adopted it since the pandemic started, and three others who already used electronic transactions plan to increase their use among customers. As a result, four businesses have found it easier to track their finances since the pandemic started.

## 5. Discussion

Our study contributes to research and practice in several ways. We focus on the overall implications of eCommerce in an SME organization. We opted to generalize the recommendations on specific functions such as operation, sales and marketing, and finance.

### 5.1. Theoretical Implications

We sought to determine the pandemic's disruptive role on the impact of eCommerce on significant SME functions. Our study participants are smaller organizations with limited resources in terms of the people, technology, and processes traditionally required to compete with their larger competitors. The results of our research show that most of these organizations use eCommerce platforms—either through their company website or by integrating an eCommerce platform—primarily for sales and marketing, despite the potential advantages of using eCommerce in their operations and finance areas to compensate for their relative lack of resources.

As a result, these organizations outsource processes in these areas to other providers or implement them using other systems. Such systems are typically disintegrated from the eCommerce system and thus require the SMEs to manually combine data from fragmented processes to obtain a complete view of their organizational requirements. The pandemic has reinforced this tendency to outsource non-core functions, while adopting digital tools to maximize the returns.

Further, while eCommerce enabled these SMEs to compete in a wider geographic area, resource shortages still handicap their ability to scale to support the growth required in the international market. This was especially true during the pandemic, which brought disruption to supply chains and human resources. Our findings solidify the constraints identified in the literature regarding how scarcity of resources limits SMEs in their ability to expand.

In sales and marketing, we found that the data SMEs gathered from their eCommerce platforms loosely define their short-term marketing strategies to advertise products through digital ads and social media. Few of the SMEs used metrics to selectively target their marketing approaches. Instead, most of our respondents used the data to better understand their customers. Indeed, the SMEs often used website features aimed at ensuring customer satisfaction to gather customer feedback, questions, and complaints rather than to improve their product quality. They also often failed to track customer retention and brand loyalty by leveraging data analytics. These proved detrimental during the pandemic as SMEs struggled to determine which market segments to prioritize.

In terms of financial management functions, SMEs expected their marketing costs to be offset by using eCommerce. Only a few respondents were able to reduce their transaction costs using eCommerce. Working capital may be critical to eCommerce adoption, but the results are not instantly realized, and several SMEs are considering changes in this area based on their experiences thus far. Further, most SMEs do not have immediately available resources to design, improve, and analyze their website content. As a result, even when the recent pandemic lockdowns improved their revenues, some identified eCommerce adoption as a cost increase rather than a reduction in relation to such gains. In fact, eCommerce was an effective driver of sales growth, both through advertising and in its ability to allow the SMEs to instantly increase their prices as supply chain costs rose. Increased eCommerce visibility in financial management may show long-term gains, but currently, the ineffective use of fragmented systems masks the benefits of eCommerce in financial monitoring.

In terms of operations, most of the SMEs claim that eCommerce improved their business efficiencies upon implementation and supported the businesses during the pandemic. It also reduced the need for additional resources and unnecessary process steps within the organization. Instead, using the platform allowed the SMEs to outsource functions or use various other systems to increase competitiveness. Using the technology to effectively communicate with customers was highly cited by SMEs as a key advantage that supported operational functions; indeed, this communication flow was seen as the primary driver of sales in the SME supply chain processes.

Given the scarcity of resources in most SMEs, an important contribution of our research is in testing the actor network theory (ANT). The study's contribution to theory is in the critical application of ANT to SMEs, emphasizing the inclusion of non-human actors (effective technologies), that give the organizations data needed to offset their relative lack of human resources. Indeed, the use of technology and outsourcing resources should align with defined organizational roles aimed at meeting the business owner's objectives. That is, SMEs should "hire" technology, being the eCommerce platform and integration of other systems into this platform, to address "jobs to be done" to replace human actors within their organizations.

### 5.2. Managerial Implications

ANT requires that SME managers or decision makers (typically, the business owners) let their eCommerce website guide their decision making. Deploying eCommerce is most useful when SMEs integrate data from various information sources into the eCommerce platform to support three key business areas in the organization.

Sales and marketing are the area in which SMEs most use eCommerce. However, the eCommerce website must be utilized beyond digital advertising, taking advantage of its value as an instant communication mechanism with customers. These customer communications can be used to better inform SMEs' future decision making, improve customer service, and enhance product quality. Further, while using the platform to penetrate global markets will require additional resources, SEO can provide profiles of target areas to focus on for international expansion, as well as identify untapped markets. This compounds the knowledge required by the SME and reduces the cost of penetrating an undetermined geographical market.

In terms of finances, the SMEs use eCommerce platforms to process payments for online orders. However, fragmentation among operational functions disabled an important eCommerce functionality: linking the financial information gathered to the SMEs financial decision making, while eliminating the need for human intervention. Compared to larger organizations, SMEs face constraints on both human and financial resources. They can, however, better leverage their use of eCommerce technology to improve both financial management and asset monitoring.

Operationally, SMEs can better leverage their eCommerce websites to improve business efficiencies in several ways. First, they can better leverage eCommerce to more effectively communicate with stakeholders (customers, vendors, and employees). SMEs in our study were not yet using features in their various platforms that could improve internal and external communications. Such features, which include inventory management, sourcing, and inbound system monitoring, can simplify information flow and provide transparency, thereby offering instant, data-driven decision making. Second, many of the SMEs had fragmented systems; this requires additional resources—for manual data collection, analysis, and interpretation—if the data from these systems are to be effectively integrated with data from the eCommerce platform. Although setting up a system to automate data collection, analysis, and interpretation requires resources, organizations will benefit in terms of the resulting speed and quality of information flow. eCommerce allows automation, which reduces intermediaries that equate to higher costs. Technology and process automation may translate to cost reductions that compensate for the set-up fees and maintenance of a highly integrated system. Finally, our SMEs rarely used eCommerce in their supply chain processes. They primarily used their link to third-party vendors for downstream processes and order fulfilment. However, this order management and processing does not offer visibility into inventory, which can be useful for inventory control and order fulfillment. To achieve this, SMEs can integrate their eCommerce systems with the supply chain providers on both upstream and downstream processes.

The SMEs claimed to have recognized the value of integrating fragmented processes into the eCommerce platform and exhausting the use of technology during a disruption. Most of them acknowledged its utility to augment the lack of manpower in their organizations during the pandemic. They also identified that eCommerce cut out middlemen such as distributors and retailers even prior to the pandemic. Ironically, most organizations do not have concrete plans and programs to exploit the functionalities of an eCommerce site despite recognizing the positive impact on their organization during the pandemic.

Although a few of the respondents are considering integration of operational processes, the scarcity of resources and technical knowledge still creates a challenge for these firms to fully embrace the change. Reduction in costs as an indirect impact of the use of eCommerce in managing their supply chain—through inventory management, process optimization, coordination, and internal communication—is a secondary consideration or even disregarded.

Despite the use of eCommerce primarily for marketing their products to increase revenue, other avenues using digital advertising and SEO are not fully optimized. The major role of social media limits the SME's strategy to brand awareness, which also creates the challenge that this medium potentially drives sales to their competition.

In finance, eCommerce is utilized solely for payment processing of eCommerce transactions. SMEs are unlikely to use the platform to link eCommerce to other financial platforms used for tracking, reporting, asset monitoring, and cost reduction. Most SMEs have not fully realized the improvement of working capital position in using eCommerce.

Smaller organizations are usually apprehensive about integrating processes due to their reliance on other partners and legacy processes. While this seems to be a necessary solution due to their current size, it inhibits their ability to scale, which is fundamental to their growth. Leveraging technology eliminates the need for human intervention and creates agility. The current status quo in running business operations using manual and disintegrated processes will not be sustainable in the long run.

*5.3. Limitations and Recommendations for Future Research*

We limited the research to US-based SMEs with fewer than 70 employees, with at least two of similar organizational size. In so doing, we can compare implications for groups of organizations based on their sizes in contrast with SMEs with more employees. Future researchers may expand the research to larger SMEs with more than 70 employees, since, per US definition, any organization with fewer than 500 employees is considered an SME. Further study may be used to compare micro-SMEs, or organizations with fewer than 10 employees, or it may be directed towards medium-sized organizations.

The scope of the research is within the US, to distinguish implications of the pandemic. COVID-19 has different restrictions in every country, thus will challenge the validity of the process and ground the time, to define BEFORE and DURING the pandemic. We also acknowledge that the effect may vary from one country to another, that limiting the cases to US will be a better comparison among SMEs.

Albeit a constraint in some countries, particularly those with limited access to the internet, the continued growth of mobile services globally enables eCommerce as a method of trade. With the emergence of mobile commerce, most technologies are available through mobile applications and websites accessible through mobile phones. These sites have the functionality to develop links to operations, sales and marketing, and finance functions for an SME in any part of the world. SMEs in developing countries such as Pakistan, Nigeria, and Malaysia, for example, are comfortably familiar with the use of mobile phones more than computers and the internet.

As our research shows, the relationship between SME success and eCommerce capabilities is in early stages and warrants further study. Both SMEs and eCommerce are set to evolve rapidly over the next decade as organizational systems become more decentralized, with a bias toward smaller organizations and improved responsiveness. eCommerce is also evolving new innovations in experiences, transactions, and customer relationships. Further, the physical infrastructure—particularly in supply chain processes such as logistics—is mutating into a form that can better serve the needs of both SMEs and eCommerce platforms. This will only drive increased adoption until digital operations are the only option for eCommerce businesses. While the eCommerce platform is a vital actor in an SME's network, the business owner, as the obligatory passage point (OPP) must take advantage of eCommerce's utility in operations and finance, and not just in sales and marketing, where the benefit is more apparent. Integration of most, if not all of the systems might incur additional cost on the front end, but it will save costs in the long run. Leveraging the use of technology, given the scarcity of SME resources, improves business efficiencies, reduces manual processes, and increases the bottom line.

The initial proposal for this research included post-pandemic research, with the hopes that the situation normalizes during our data gathering phase by mid-2021. Our intent is to realize the full impact of eCommerce during and after a disruption. The continuous development of the virus disallowed our ability to provide recommendations learned during the pandemic that may be replicated when the normal business activities continue. Although some of the SMEs in this study identified post-pandemic plans, we were not able to validate the efficacy of eCommerce in this business state. We recommend that future researchers explore how SMEs utilized post-pandemic learnings that may be adopted to avert future disruptions.

Destructive, natural, and man-made disruptions are also likely to occur more often. A major force behind the world's ability to successfully function during the COVID-19 pandemic has been the unprecedented interconnectedness of the world in this age. Systems integration among all actors within a network—in this case, the SME—will be critical to sustaining businesses during disruptions. Physically, goods and services are being used more communally than ever before. This is enough proof that the world is getting smaller. Goods are sourced as raw materials in certain parts of the world and sold as finished products in another. Online, the close knitting of information, financial, and operational systems is driving the absorption of shocks in different parts of the world, for both upstream

and downstream processes. As they face the future, it is therefore imperative for SMEs to master their best chance at responsiveness and continued survival: eCommerce.

**Author Contributions:** Conceptualization, R.B., M.G.S., O.T. and S.S.; methodology, R.B., M.G.S., O.T. and S.S.; validation, R.B., M.G.S. and O.T.; formal analysis, R.B., M.G.S., O.T. and S.S.; investigation, R.B., M.G.S. and O.T.; resources, R.B., M.G.S. and O.T.; data curation, R.B., M.G.S. and O.T.; supervision, S.S.; writing—original draft preparation, R.B., M.G.S. and O.T.; writing—review and editing, R.B., M.G.S., O.T. and S.S.; visualization, R.B., M.G.S., O.T. and S.S. All authors have read and agreed to the published version of the manuscript.

**Funding:** This research received no external funding.

**Institutional Review Board Statement:** The study was conducted in accordance with the Declaration of Helsinki, and approved by the Institutional Review Board of Georgia State University (protocol code H21681 and approved on 07/09/2021).

**Informed Consent Statement:** Informed consent was obtained from all subjects involved in the study.

**Data Availability Statement:** Data available on request due to privacy/ethical restrictions.

**Acknowledgments:** The authors would like to thank the J. Mack College of Business at Georgia State University (GSU) for guidance, encouragement, advice, and unlimited support.

**Conflicts of Interest:** The authors declare no conflict of interest.

## Appendix A

**Table A1.** Business area, focus, and impact of eCommerce on SMEs.

| Business Area | Focus | eCommerce Impact |
|---|---|---|
| Operations | Processes | Business efficiencies [18,19] |
| | | Use of available resources [20] |
| | | Disintermediation [21] |
| | Technology | Information exchange and transparency [2,23] |
| | | Information flow more efficient [19,22] |
| | Supply Chain | Streamlined supply chain [2,25] |
| | | Simplifies international transactions [2,25,30,48] |
| Sales and Marketing | Marketing | Creation of Marketing Strategies [4,28] |
| | | Cost of Advertising [28] |
| | | Use of Social Media [28,35] |
| | | Source of Marketing Data [27,30] |
| | Globalization and New Market Entry | Penetrates remote markets [2] |
| | | Competes in a global market [3,31] |
| | | Marketing channel that reaches global customers [32] |
| | | Immediate accessibility to customers [24,27] |
| | Customer Service | Reduces dependence on distributors for customer engagement [2,33] |
| | | Customer feedback to enhance product quality [11,31] |
| | | Customer Retention and Brand Loyalty [28,34,36] |
| Finance | Revenue Growth and Cost Reduction | Higher ROI on Marketing Cost [49] |
| | | eCommerce is significantly effective on SME performance, financial performance, internal process, customers, growth and learning [50] |
| | | Lower transaction costs [51] |
| | Improved Working Capital | Disintermediation reduces transaction costs and value leakage in the value chain [2] |
| | | Reduced Operating Costs [52] |
| | Financial Management and Asset Monitoring | eCommerce reduces the cost of monitoring business and financial operations [20] |
| | | Among the tangible benefits is the reduced production cost [19] |
| | | eCommerce reduces transaction and coordination costs through interconnective networks and online databases globally [51] |

## Appendix B

**Table A2.** Semi-structured Interview Questions per Impact of eCommerce to an SME.

| Impact of eCommerce | Area | Impact | Questions |
|---|---|---|---|
| Operations | Processes | Business efficiencies | How did the integration of these business functions improve your operations? Were there changes during the pandemic? Are you planning to make any changes to these functions once the pandemic is completely over? |
| | | Use of available resources | How did the use of eCommerce impact other organizational functions i.e., sales and marketing, or finance? Which business functions are not integrated? What challenges did you experience on functions that are disintegrated? |
| | | Disintermediation | When you started to use eCommerce, were there any middlemen eliminated in any of your processes? Were there any changes during the pandemic? Are you anticipating any changes after the pandemic? |
| | Technology | Information exchange and transparency | How do you use the eCommerce platform to communicate with your customers? employees? and suppliers? Were there any changes during the pandemic? Were your customers able to better find information about your products? How do you receive feedback from customers? |
| | | Information flow more efficient | Prior to the pandemic, are there any digital steps or tools in your information flow? Please describe the process. For activities that are disintegrated to the eCommerce platform, how do you gain visibility to your data? Was there any change during the pandemic? Are you anticipating any changes when the pandemic is completely over? On the information flow using eCommerce, did you see an improvement in quality and speed of information flow? Which function improved? Which function worsened? Was there any change during the pandemic? Are you anticipating changes after the pandemic? |
| | Supply Chain | Streamlined supply chain | Please describe how your eCommerce is linked to your supply chain. Were there any changes during the pandemic? Are you making changes to the supply chain process after the pandemic? |
| | | Simplifies international transactions | What is the percentage of international and local sales? What is your normal sales process for domestic sales? Is it different from transactions coming from international sales transactions? What is the difference? |
| Sales and Marketing | Marketing | Creation of Marketing Strategies | What role does your eCommerce platform play in your digital marketing activities? Were there any changes during the pandemic? Are you implementing new marketing strategies when the pandemic is over? |
| | | Cost of Advertising | Does your business run digital ads? Why or why not? If you are, was this implemented before the pandemic? Will there be changes after the pandemic? Is your website utilized for SEO? If not, why? Was this utilized before the pandemic? Did you make changes on your SEO utilization during the pandemic? Are you planning on changing your SEO utilization after the pandemic? |
| | | Use of Social Media | Are you using social media to advertise/market your products? Which one? Have you used this before the pandemic? What improvements did it contribute to your business? What changes in the use of social media did you implement during the pandemic? Are you planning on making other changes once the pandemic is over, if so, please describe what changes will you be implementing? |
| | | Source of Marketing Data | Does your website offer customized content such as customer profile, country of origin or other metrics? Was this implemented before the pandemic? How did it help? Was customization more useful during the pandemic? Are you planning to make changes in the customization after the pandemic? |

**Table A2.** *Cont.*

| Impact of eCommerce | Area | Impact | Questions |
|---|---|---|---|
| Sales and Marketing | Globalization and New Market Entry | Penetrates remote markets | Describe your market before using eCommerce. Did you market expand after using eCommerce, were there new areas that you distribute to?<br>How did this change during the pandemic?<br>Are you planning any changes to your distribution once the pandemic is over? |
| | | Competes in a global market | How did eCommerce allow you to compete in the global market?<br>Did the pandemic impact how you compete with the global market?<br>What changes will you implement on your global expansion when the pandemic is over? |
| | | Marketing channel that reaches global customers | What are your major international markets? How did your eCommerce platform expand your market internationally?<br>Was there a change during the pandemic?<br>Are you expanding further after the pandemic? |
| | Customer Service | Immediate accessibility to customers | How do customers request for additional information on your products? Do they do it through your eCommerce platform or marketplaces? Has this always been the process before the pandemic?<br>What changed in your customer communication during the pandemic? Are you still using the same process? If not, what changed? Are you planning to make changes after? |
| | | Reduces dependence on distributors for customer engagement | Other than eCommerce, what are other ways of selling your product before the pandemic (do you use retailers, distributors etc.)?<br>Was there any change during the pandemic?<br>Are you anticipating changing this model when the pandemic is over? |
| | | Customer feedback to enhance product quality | How do you receive feedback from your customers, is it through the platform? If not, what other means do you provide feedback? Do you use the feedback received from the customers? How so?<br>Was there any change in the feedback process during the pandemic?<br>Are you planning to make any changes to this feedback process once the pandemic is over? |
| | | Customer Retention and Brand Loyalty | How do you track returning customers or customer loyalty? Does your platform enable you to do that?<br>Was there any change during the pandemic? Please elaborate.<br>Are you making changes when the pandemic is over?<br>Because of eCommerce, did you see an increase in the number of regular customers before the pandemic?<br>What happened to the regular customers during the pandemic?<br>What do you think will be the outcome on return business after the pandemic? |
| Finance | Revenue Growth and Cost Reduction | Higher ROI on Marketing Cost | How did your marketing costs change after the implementation of eCommerce? Did it increase or decrease?<br>How did this change during the pandemic?<br>Are you going make any changes after the pandemic- will this reduce marketing cost or increase it? |
| | | eCommerce is significantly effective on SME revenue performance, customers, growth | How did the use of the eCommerce platform contribute to the revenue growth of your company before the pandemic?<br>Was there any change during the pandemic? Please explain.<br>How do you forecast your business growth when the pandemic is over?<br>Did the price for your major products increase or decrease because of the use of eCommerce or marketplaces?<br>Did this change during the pandemic? What caused the change?<br>Are you planning for other price changes after the pandemic?<br>How did eCommerce impact your ability to change prices before the pandemic?<br>Did the pandemic change your ability to change prices? How?<br>Are you planning to make changes after the pandemic? |

**Table A2.** *Cont.*

| Impact of eCommerce | Area | Impact | Questions |
|---|---|---|---|
| Finance | Improved Working Capital Position | Lower transaction costs | How does your eCommerce platform support your sales transactions? Did this reduce your transaction cost? Did this transaction cost change during the pandemic? Are you expecting this to change when the pandemic is over? |
| | | Disintermediation reduces transaction costs and value leakage in the value chain | Do you use new systems/tools that you did not have before the pandemic? If any, which ones were added? Were there changes during the pandemic? Are you planning to add more tools after the pandemic is over? |
| | | Reduced Operating Costs | Did you realize any cost reduction after the implementation of eCommerce? Can you enumerate which costs were reduced? Was there any change to this during the pandemic? Are you creating cost reduction measures after the pandemic? |
| | Financial Management and Asset Monitoring | Reduces the cost of monitoring business and financial operations | Do you track the effectiveness of your marketing activities on eCommerce, marketplaces or social media? If so, has it shown effectiveness before the pandemic? How about while the pandemic is going on? Do you intend to make changes when the pandemic is over? How so? Have you found it easier to track your finances and transactions since you implemented eCommerce? Was there a change on how you track finances during the pandemic? Are you going to implement changes in tracking your finances after the pandemic? |
| | | Reduced production cost | How did you handle transactions with your customers and suppliers before the pandemic? Is it through your eCommerce platform? Did this change during the pandemic, how? What changes are you planning to implement when the pandemic is over? |
| | | Reduces transaction and coordination costs | What type of payments do you accept—cash or electronic? If electronic, what type(s) of payment? What is the percentage between digital and electronic transactions? Had there been any change since the pandemic started? Do you project any changes after the pandemic? |

## Appendix C

**Table A3.** Illustration of Scoring Progression for Questions 1 and 2.

| Interpretation | Case | Author Scores | Case Aggregate Scores | Average Question Score | Conclusion |
|---|---|---|---|---|---|
| Question 1 | Case 1 | Author 1 Author 2 Author 3 | CAS 1 = Author 1 + Author 2 + Author 3 | (CAS 1 + CAS 2 + CAS 3)/3 | Conclusion Question 1 |
| | Case 2 | Author 1 Author 2 Author 3 | CAS 2 = Author 1 + Author 2 + Author 3 | (CAS 1 + CAS 2 + CAS 3)/3 | Conclusion Question 1 |
| | Case 3 | Author 1 Author 2 Author 3 | CAS 3 = Author 1 + Author 2 + Author 3 | (CAS 1 + CAS 2 + CAS 3)/3 | Conclusion Question 1 |
| Question 2 | Case 1 | Author 1 Author 2 Author 3 | CAS 1 = Author 1 + Author 2 + Author 3 | (CAS 1 + CAS 2 + CAS 3)/3 | Conclusion Question 2 |
| | Case 2 | Author 1 Author 2 Author 3 | CAS 1 = Author 1 + Author 2 + Author 3 | (CAS 1 + CAS 2 + CAS 3)/3 | Conclusion Question 2 |
| | Case 3 | Author 1 Author 2 Author 3 | CAS 1 = Author 1 + Author 2 + Author 3 | (CAS 1 + CAS 2 + CAS 3)/3 | Conclusion Question 2 |

**Table A4.** Scoring Table for questions 4–6.

| Question | 4. Tell Us about Your Products- What Are the Top 3 Products | 5. What eCommerce Platform Do You Use | 6 a. Why You Choose Your eCommerce Platform What Are the Benefits b. How Long Have You Used This Platform |
|---|---|---|---|
| CASE 1 | | | |
| Author 1 Score | 1 | 1 | 1 |
| Author 2 Score | 1 | 0 | 0 |
| Author 3 Score | 1 | 1 | 1 |
| Aggregate Score | 3 | 2 | 2 |
| CASE 2 | | | |
| Author 1 Score | 1 | 1 | 1 |
| Author 2 Score | 0 | 1 | 1 |
| Author 3 Score | 0 | 1 | 1 |
| Aggregate Score | 1 | 3 | 3 |
| CASE 3 | | | |
| Author 1 Score | 1 | 1 | 0 |
| Author 2 Score | 1 | 1 | 0 |
| Author 3 Score | 1 | 1 | 0 |
| Aggregate Score | 3 | 3 | 0 |
| CASE 4 | | | |
| Author 1 Score | 0 | 1 | 1 |
| Author 2 Score | 0 | 1 | 0 |

**Table A4.** *Cont.*

| Question | 4. Tell Us about Your Products- What Are the Top 3 Products | 5. What eCommerce Platform Do You Use | 6 a. Why You Choose Your eCommerce Platform What Are the Benefits b. How Long Have You Used This Platform |
|---|---|---|---|
| Author 3 Score | 0 | 1 | 1 |
| Aggregate Score | 0 | 3 | 2 |
| CASE 5 | | | |
| Author 1 Score | 0 | 1 | 0 |
| Author 2 Score | 0 | 1 | 0 |
| Author 3 Score | 0 | 1 | 0 |
| Aggregate Score | 0 | 3 | 0 |
| CASE 6 | | | |
| Author 1 Score | 1 | 1 | 1 |
| Author 2 Score | 1 | 1 | 0 |
| Author 3 Score | 1 | 0 | 1 |
| Aggregate Score | 3 | 2 | 2 |
| CASE 7 | | | |
| Author 1 Score | 0 | 1 | 0 |
| Author 2 Score | 0 | 0 | 1 |
| Author 3 Score | 0 | 1 | 1 |
| Aggregate Score | 0 | 2 | 2 |
| CASE 8 | | | |
| Author 1 Score | 0 | 1 | 1 |
| Author 2 Score | 0 | 1 | 1 |
| Author 3 Score | 0 | 1 | 1 |
| Aggregate Score | 0 | 3 | 3 |
| Aggregate Scores | | | |
| Case 1 | 3 | 2 | 2 |
| Case 2 | 1 | 3 | 3 |
| Case 3 | 3 | 3 | 0 |
| Case 4 | 0 | 3 | 2 |
| Case 5 | 0 | 3 | 0 |
| Case 6 | 3 | 2 | 2 |
| Case 7 | 0 | 2 | 2 |
| Case 8 | 0 | 3 | 3 |
| Aggregate Average Scores | | | |
| Aggregate Average | 1.25 | 2.625 | 1.75 |
| Supports | 4 | 8 | 6 |
| Rejects | 0 | 0 | 0 |
| Conclusion | 0 | Supported | Supported |
| | 0 | 0 | 0 |
| | Inconclusive | 0 | 0 |

**Appendix D**

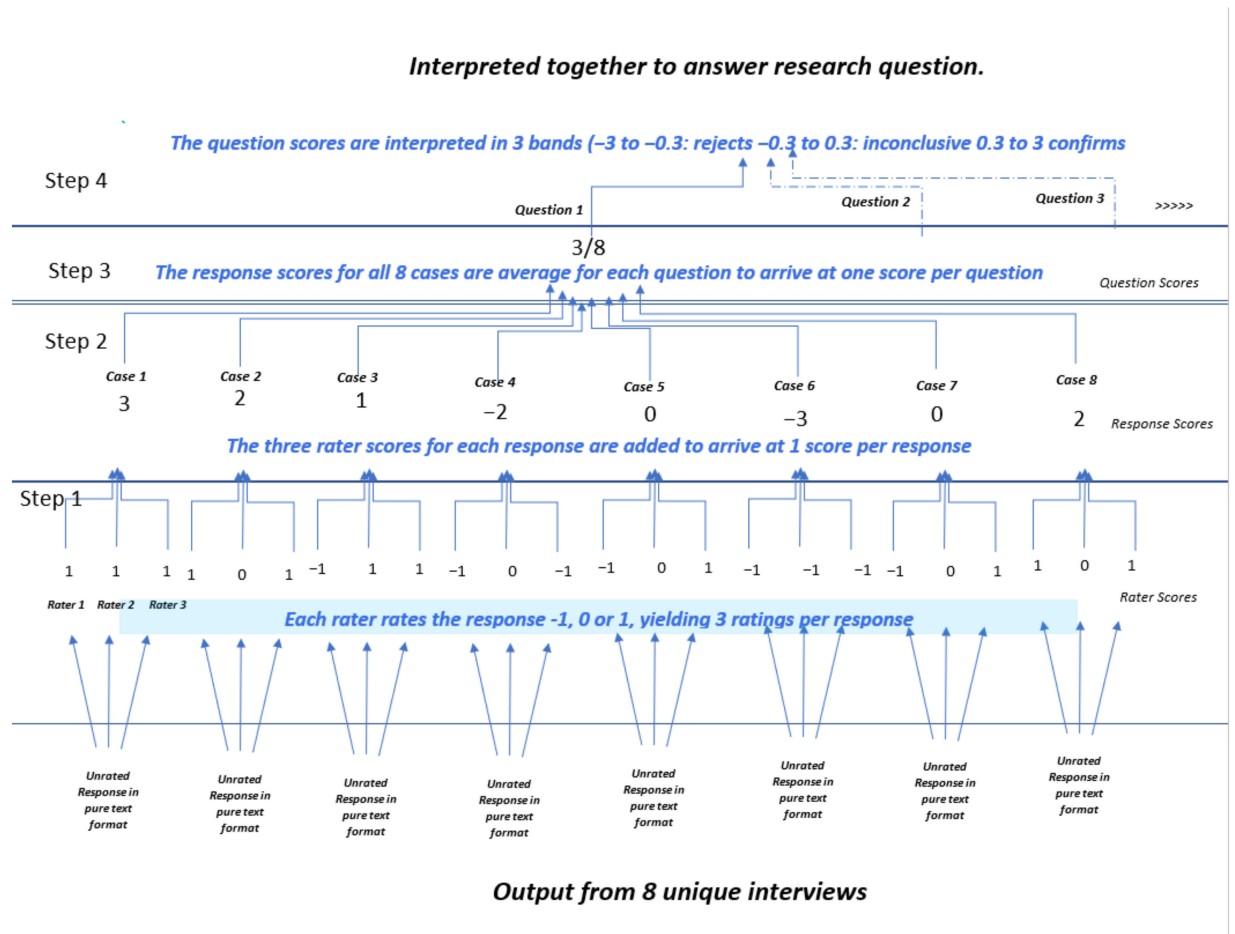

**Figure A1.** Four step scoring process.

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
