# Peer review of "How Does a Pandemic Disrupt the Benefits of eCommerce? A Case Study of Small and Medium Enterprises in the US"

_jtaer, doi:10.3390/jtaer17020028_

Round 1
Reviewer 1 Report
This is a very interesting topic to study in. The relevant literature has been reivewed and the research is properly designed. It would be better if the discussion on the results and scoring can be explained a bit futher. Some citations are missing and shown as Error in the manuscript - please try to use the property citation and referencing. The article is concluded well.
Reviewer 2 Report
The paper is based on interesting topic. However, I feel that authors could further improve the paper. I draw here some indications to do so:
- Introduction--Here make motivation clear. Use more empirical backing.
- Include which guidelines you have followed to conduct this research. Further, mention something about your empirical research protocol, its execution and then focus on how you addressed threats to validity.
- Implications are perfectly mentioned. But support of provide contradictory views by adding more empirical evidences.
Reviewer 3 Report
The aim of the paper was to investigate the benefits of using ecommerce platforms for small and medium-sized companies, in the case of US companies.
The paper is interesting, but there are still areas for improvement.
The period in which the interviews were conducted must be added to the methodology section. It is necessary to explain more clearly how the eight companies were chosen. The US is a big country, why were the eight and not others? What would be the selection criteria, apart from those mentioned?
In Discussions and Conclusions it is necessary to mention how the data can be generalized for companies from other countries, for example those with limited internet access.
Round 2
Reviewer 2 Report
Author had addressed all my concerns.
Reviewer 3 Report
The paper is improved with all my suggestions.